# Social exclusion concepts, measurement, and a global estimate

**Jose Cuesta[1], Borja López-Noval[2], Miguel Niño-Zarazúa**  **[3]** *

**1** Social Sustainability and Inclusion Global Practice, The World Bank, Washington, D.C., United States of America, **2** Department of Economics and Statistics, Universidad de León, León, Spain, **3** Department of Economics, SOAS University of London, London, United Kingdom

* mn39@soas.ac.uk

## Abstract

Multiple estimates exist of global monetary and multidimensional poverty, but populations at risk of social exclusion still lack a worldwide estimate. This paper fills this gap by providing the first estimates of the share and number of populations at risk of social exclusion worldwide. The paper contributes to the literature in three important respects. First, it develops a conceptual framework of social exclusion that emphasizes the relative, multidimensional, and dynamic features of exclusion. Second, it proposes a macro-counting methodology that allows measuring populations at risk of exclusion based on identity, circumstances, and socioeconomic conditions, while advancing a protocol to avoid double counting of individuals at risk of social exclusion. Third, the empirical strategy provides to the best of our knowledge, the first estimates of populations at risk of social exclusion by dimensions of exclusion on a global and regional scale. Overall, we estimate that between 2.33 and 2.43 billion people—roughly 32 per cent of the global population—are at risk of social exclusion. The South Asia and East Asia and Pacific regions contain 1.3 billion such people, with India and China alone home to 840 million of them. Meanwhile, 52 per cent of sub-Saharan Africa's population is vulnerable to exclusion, the greatest share of any region. Our findings have important policy implications. While antipoverty policies can support household consumption and smooth its volatility among the poor, they are unlikely to address social exclusion stemming from ethnic, racial, or gender discrimination. Therefore, addressing exclusion necessitates a suite of multiple interventions tailored to distinct groups and sustained over time.

## 1. Introduction

Despite the mounting evidence around certain populations that are at risk of being socially excluded, there has been no systematic and comprehensive attempt to estimate their number at global and regional level. Global estimates for specific populations at risk of exclusion do exist but are rarely frequent or systematic. For example, the United Nations High Commissioner for Refugees (UNHCR) and the World Health Organization (WHO) report annual numbers for forcibly displaced populations and victims of GBV, respectively. However, global estimates of persons with disabilities are highly imprecise, while there are no such estimates of

**Data Availability Statement:** All relevant data are within the manuscript and its Supporting Information files.

**Funding:** This study was supported by the World Bank. The funder had no role in study design, data collection and analysis, decision to publish, or

preparation of the manuscript. One author, Jose Cuesta, receives a salary from the funder.

**Competing interests:** The authors have declared that no competing interests exist.

lesbian, gay, bisexual, transgender and intersex (LGBTI) people. This contrasts with the numerous global income poverty estimates, global multidimensional poverty estimates, and global child poverty estimates [1, 2].

The absence of global estimates of populations at risk of social exclusion has contributed to the interchangeable use of poverty and social exclusion in some contexts [3]. This is problematic from both a conceptual and policy perspective to the extent that significant shares of socially excluded populations are nonpoor. Without aggregated and integrated estimates, it is not possible to monitor how crises, public policies, or demographic changes impact social exclusion. Furthermore, antipoverty policies are unlikely to effectively address social exclusion stemming from ethnic or racial discrimination. Similarly, policies aimed at raising minimum wages may impact the working poor but have no effect on exclusion caused by the stigma associated with sexual orientation.

This paper contributes to the literature by providing the first estimates of people at risk of social exclusion worldwide and compares those estimates with monetary and multidimensional poverty numbers. This comparison provides insights into the limitations of anti-poverty policies to tackle the root causes of social exclusion and calls for complementary policy interventions. The proposed methodology lays the foundation for measuring populations at risk of social exclusion globally and contributes to international and national efforts in building monitoring systems and formulating effective policy responses.

In order to implement our empirical strategy, we have developed a conceptual framework of social exclusion that builds on Sen's [4] capability approach, and designed an empirical strategy to estimate the shares of populations that are at risk of social exclusion based on a macro counting approach. Our conceptual framework places emphasis on the relative, multidimensional, and dynamic features of exclusion, identifying specific groups that are particularly vulnerable to exclusion based on identity, circumstances, and socioeconomic conditions. This approach benefits from theories of social exclusion that focus on notions of social exclusion as a lack of participation in society with dignity [4–6], and more recent studies, which highlight processes and equity considerations [7–9]. Our empirical strategy draws on the most credible data sources estimating vulnerable population groups across countries and develops a protocol to minimize double counting. The proposed methodology provides a framework for measuring and monitoring social exclusion, facilitating the analysis of trends and potential sources of exclusion. To the best of our knowledge, we are presenting the first global and regional estimates of people at risk of social exclusion based on a sound conceptual framework and a rigorous methodology.

Overall, we find that between 2.33 and 2.43 billion people, or between 31.1 and 32.4 percent of the global population, are at risk of social exclusion based on identity, circumstances, and socioeconomic conditions. The results underscore the following policy implications: first, anti-poverty policies are likely to overlook approximately 1.5 billion people who although are nonpoor, are nonetheless at risk of exclusion. Second, addressing social exclusion requires multiple and differentiated interventions across contexts and sustained over time. Third, tackling social exclusion requires precise estimates, rigorous methodologies, and data to monitor its incidence, severity and sources. The remainder of the paper is structured as follows. Section 2 reviews the literature and presents our conceptual framework. Section 3 describes the methodological approach. Section 4 discusses the sources of data used; Section 5 presents the estimates of populations at risk of exclusion, both globally and regionally, while Section 6 concludes.

## 2. Conceptualizing social exclusion

The concept of social exclusion and its multiple manifestations and drivers have been the focus of a growing literature [4–6, 10]. [11] defines social exclusion as "*a state in which*

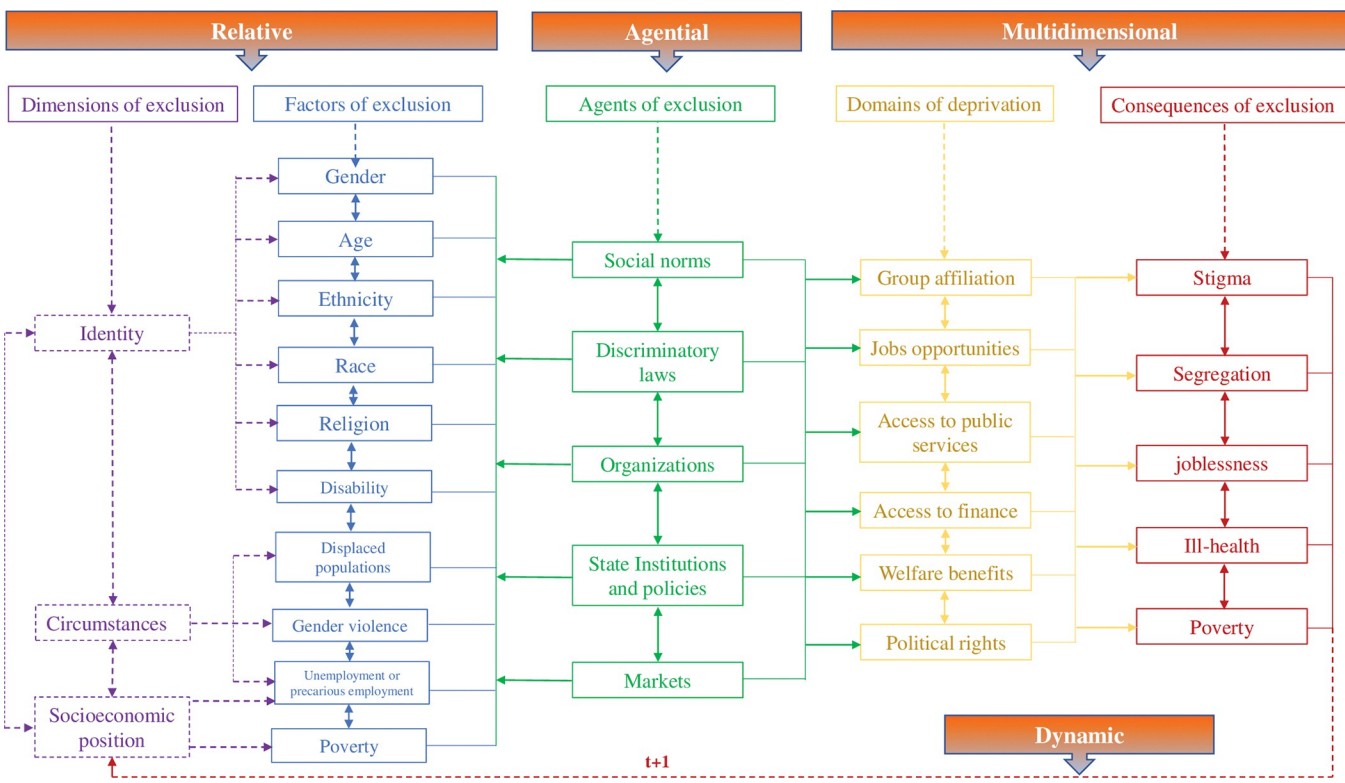

**Fig 1. Conceptual framework of social exclusion.** Source: Authors.

*individuals are unable to participate fully in economic, social, political and cultural life, as well as the process leading to and sustaining such a state.*" [8, 12] emphasize the relative processual nature of exclusion, underscoring the terms in which individuals and social groups are excluded from taking part in society. [13, 14] more explicitly focus on the presence of discriminatory practices such as racism, xenophobia, and ageism that hinder individuals' participation in economic, social, and political life. [9] also emphasize the implications of exclusion in terms of lack of access to markets and services, as well as limited political, cultural and social participation. Thus, the deprivations that emerge from social exclusion vary widely, although they result from three overarching dimensions, as illustrated on the left-hand side of Fig 1.

First, people are at risk of exclusion because their *identities* diverge from established norms and customs. These identities can reflect gender, age, race, caste, and ethnic characteristics, or religious and political affiliations [15–20]. This is certainly not the same as saying that all women, ethnic minorities or LGBTI populations are inevitably excluded because of their identity, but that in specific contexts, they experience high risk of exclusion.

Second, people's *circumstances*, such as being forcibly displaced because of conflict or poverty, or being a victim of gender-based violence, can leave them at risk of exclusion, especially in contexts where discriminatory norms, laws, and institutions exist [21–27].

Third, people in a *disadvantaged socioeconomic position* are more likely to be subject to exclusion because low educational attainment, unemployment, or poverty limit their opportunities to access labor, credit, and insurance markets or exercise their political rights [28–31]. These dimensions should not be—simplistically—considered sufficient nor necessary conditions for exclusion but, rather, contributing risk factors to exclusion in specific contexts and under perilous combinations of socioeconomic conditions.

While there is no consensus on the dimensions that constitute the most acute forms of social exclusion (and that should be included in global empirical estimates), there is a general agreement on its multidimensional and dynamic nature, and the relational processes by which groups are excluded. [5] points to three elements that are central to the analysis of social exclusion: 1) the *relativity* of its conceptualization, 2) *dynamics* underpinning the mechanisms of exclusion, and 3) the *agency* involved in the act of excluding others. Furthermore, [32]:9 underscore the *multi-dimensional* process of social exclusion: "*It involves the lack or denial of resources, rights, goods and services, and the inability to participate in the normal relationships and activities, available to the majority of people in a society, whether in economic, social, cultural or political arenas*".

There are different actors at work in social exclusion. Economic forces related to globalization can exclude people, as can the nation state and its institutions, and even individuals. Understanding this multidimensionality of agency involves unpacking interactions between influences and outcomes at different levels—individual, family, community, national and global [33]. Relativity, agency, dynamism and multidimensionality combined encompass, as seen in Fig 1, the dimensions, factors and agents of exclusion, and the domains of deprivations and ultimate consequences that result from these dynamics.

People are socially excluded in the context of relational interactions. For instance, people living in an affluent city may be excluded because of their identity or perceived position in that society, without being materially poor. This is a critical distinction between social exclusion and poverty, as the latter reflects absolute conditions of deprivation, a point [34]: 669 has emphasized: "*Poverty is not just a matter of being relatively poorer than others in the society, but of not having some basic opportunities of material wellbeing, the failure to have certain minimum "capabilities". The criteria of minimum capabilities are absolute, not in the sense that they must not vary from society to society, [. . .] or over time [. . .], but people's deprivations are judged absolutely and not simply in comparison with the deprivations of others in that society.*"

Exclusion is also relational in that it requires an excluding agent and someone who is excluded. This relational feature is not immediately evident in the case of poverty. Thus, the importance of social exclusion as a concept relates to its focus on *relational* identities, circumstances, and socioeconomic positions, which through the actions of others, lead to *relative* deprivations in social domains, which can ultimately manifest in stigma, segregation, joblessness, and poverty [35].

It is important to note that the intrinsically relative, multidimensional and dynamic characteristics of social exclusion distinguish it from *social inclusion*, an often interchangeably used, but not opposite equivalent concept. Social inclusion gives emphasis to individual agency, opportunity and participation under the compliance with certain norms and requirements for membership, even if such conformity can lead to social exclusion [36]. Indeed, [4] draws attention to unfavorable terms of inclusion and adverse participation in exploitative occupations that can deepen or reproduce the roots of social exclusion. [5] also points out that while unemployment may cause social exclusion, being employed does not necessarily leads to social inclusion; that would depend on the conditions of employment. Since our interest is in estimating the global scale of risk factors that emerge due to identities, circumstances and disadvantage socioeconomic positions, we focus social exclusion.

While belonging to multiple disadvantaged identities or experiencing several adverse circumstances can amplify the severity of exclusion, the conditions that underpin these exclusionary factors can change over time [21]. Political crises, disasters, conflict, economic shocks, and demographic transitions can temporarily worsen the risk of exclusion and discrimination among under-represented groups. Being at risk of social exclusion because of religious, ethnic or racial identity can be rooted in social norms that actively discriminate against people

perceived as not belonging to a community [15–17]. These norms can change over time and space (both at national and subnational levels). At the same time, exclusion is also often long-term and structural. In Latin America, indigenous populations have been historically excluded from markets and resources, and their precarious conditions continue to restrict their children from future opportunities [37]. Similarly, in South Asia, the exclusionary structure of castes has reduced intergenerational educational and occupational mobility among disadvantaged groups [38–41].

[42–45] also underscore another time-related issue when it comes to social exclusion. Relative deprivations are not fixed over time for a wide range of functionings. Thus, in order to approximate risk factors to functioning and capability failures that are also reflected in the socio-economic position of vulnerable groups, we follow [46] and adopt the concept of societal poverty line (SPL) in our empirical strategy discussed below. Unlike International Poverty Lines, the SPL relies on poverty lines that vary across and within countries over time. The SPL combines elements of absolute and relative poverty and simultaneously considers a floor that captures extreme poverty in any country, while allowing for an increase in a person's basic requirements to fulfill his or her functioning and capabilities as a country becomes richer. Refugees and displaced populations can be subject to a heightened risk of social exclusion due to discriminatory laws that *actively* deprive them of political rights or welfare benefits [22, 23]. Adverse economic conditions can also generate *passive* processes that cause long-term unemployment and exclusion. For example, in high-income countries, the rate of long-term unemployment is highly correlated with (relative) poverty rates and is a major source of social exclusion [47, 48].

In low- and middle-income countries, where long-term unemployment is generally low, it is the precarity of informal employment that passively excludes people from social protection [49]. The distinction between active and passive forms of exclusion highlighted by [4] is critical for our analysis. Understanding the roots of social exclusion is also essential when formulating policy responses.

While existing data limitations do not allow us to directly observe which specific populations experience capability failures in relevant domains of exclusion, the methodology presented in Section 3 allows us to capture these dimensions indirectly to derive global estimates of those who are at risk of social exclusion because of their identity, circumstances, and/or socioeconomic position.

## 3. Methodology

In order to estimate the size of populations at risk of exclusion globally we adopt a macro counting approach. Our method counts the number of vulnerable populations at the country level and then aggregates those counts at the regional and global level while correcting—to the extent possible—overlaps between excluded populations. This approach is conceptually simple and circumvents more methodologically complex issues at the micro level, such as defining exclusion thresholds for specific domains like markets, social services, and political spaces (see Fig 1 above).

This is a key methodological challenge in our analysis. Enduring multiple sources of exclusion can deepen the severity of relative deprivations. Being a refugee or asylum seeker as well as a member of an ethnic, racial or religious minority can severely constrain labor market participation and job prospects, irrespective of qualifications [22, 50, 51]. From a policy perspective, this intersectionality requires a broader policy space and instruments that render certain exclusionary norms "morally irrelevant." However, from a measurement point of view, it is critical to avoid double-counting multiple forms of social exclusion when generating international comparative estimates like ours.

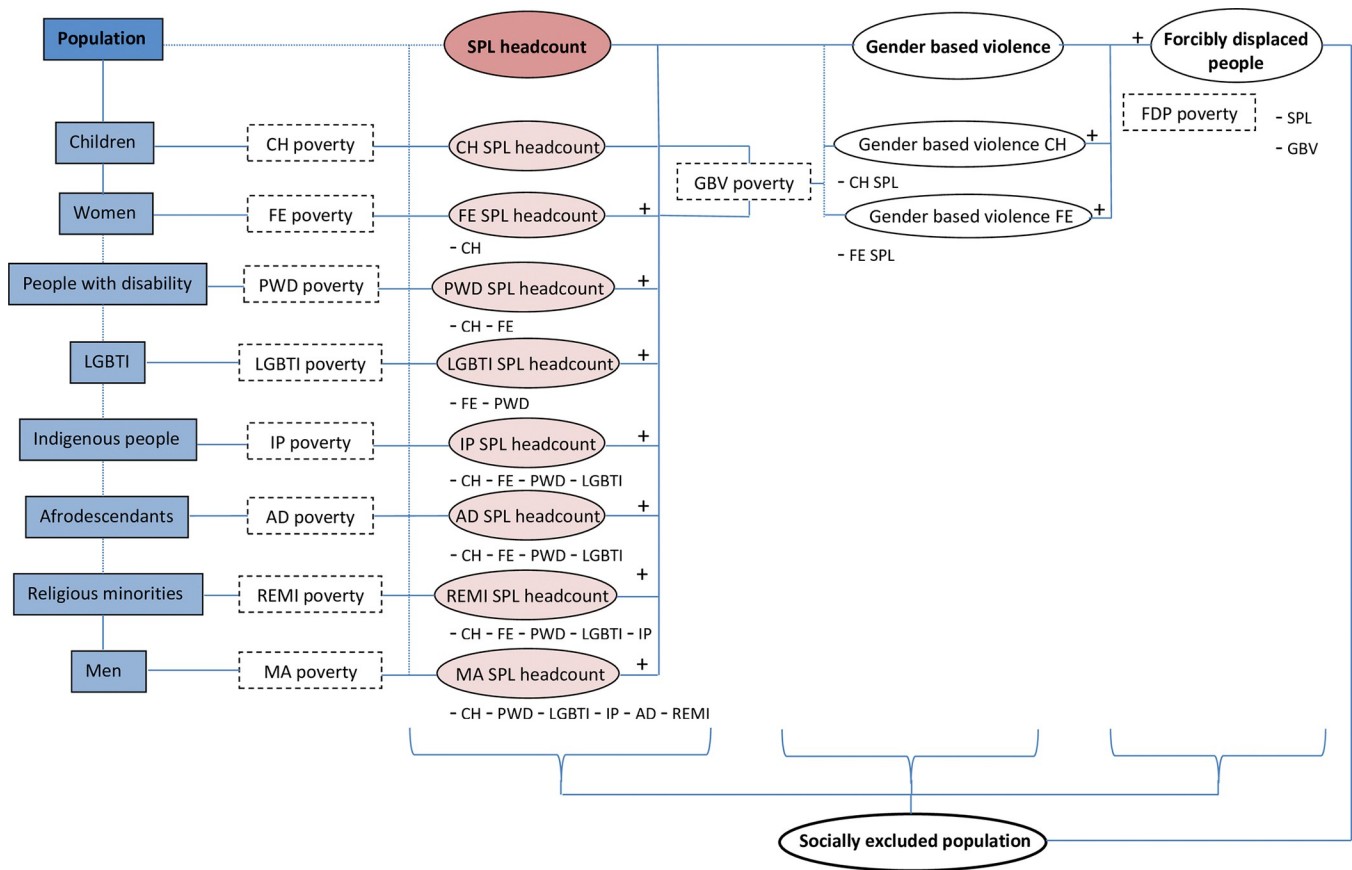

**Fig 2. A systematic approach to quantifying population groups at risk of social exclusion (after minimizing double-counting).** Source: Authors'.

Thus, to avoid double-counting, we sequentially subtract population overlaps across our predefined groups that are at risk of exclusion. Fig 2 below describes that sequential, six-step approach. The first step accounts for the total number of individuals globally that are at risk of social exclusion because of their *identities*. Drawing on multiple sources that estimate the share of the different groups at risk of social exclusion within the total population of all surveyed countries (see Section 4), we compute the total number of people who belong to a vulnerable group *g*—including people living with a disability, LGBTI people, indigenous people, Afrodescendants, and members of a religious minority—as follows:

$$pop_{gct} = pop_{ct} \times sharepop_{gct}, \tag{1}$$

where $pop_{gct}$ measures the population at risk of exclusion of group *g*, in country *c* in year *t*; $pop_{ct}$ is the total population of the country in the same year; and $sharepop_{gct}$ stands for the share of the at-risk group *g* within the total population. For example, using the estimated population of the United States in 2016, which was 323.1 million [52], and the most conservative reported estimate by the OECD [53] for the percentage of indigenous people in the US population in 2016 (1.2%), we calculate a lower bound estimate for the number of indigenous people in the United States in 2016, amounting to 3.9 million. We have absolute population data for children and women, while for other vulnerable groups, we have proportions based on gender and/or age cohorts, like in the case of people with disabilities. Thus, including population data disaggregated accordingly as in Eq (1) is straightforward.

The second step narrows the estimate of total number of people at risk of social exclusion, based on socioeconomic status for each at-risk population group. That is, it estimates the share of those living under the country's societal poverty line within each vulnerable population. Those below the SPL are expected to suffer from capability deprivations and live under conditions incompatible with a life of dignity. Unlike international or national poverty measures—which use poverty lines that remain fixed for extended periods and may not be relevant to different country income levels—societal poverty relies on a poverty line that varies across and within countries over time. As discussed above, societal poverty conveniently combines elements of absolute and relative poverty below which a life of dignity is compromised, while acknowledging that standards of living evolve over time and might change as countries become richer. A country's societal poverty line is defined as: SPL = max (US\$1.90, US\$1.00 + 0.5 × *median*), where *median* is the daily median level of income or consumption per capita [2].

A main constraint is that we lack data to determine the societal poverty headcounts for each specific population at risk of exclusion. And for other poverty measures—national or international poverty lines—we also lack consistent data across groups and countries (see Section 4 below). As a result, we estimate SPL headcount rates specific to each group ($povrate\_S\hat{P}L_{gct}$) by combining data on the overall national SPL headcount ($povrate\_SPL_{ct}$)—our preferred poverty measure, which is consistently available for most countries—with any other existing data on the poverty rate differential between each vulnerable group and the general population.

For example, we might have information on the absolute poverty rates for people with disabilities vs people with no disabilities. In that case, we use the differential of such poverty rates to generate an estimate for the number of people with disabilities below the SPL. This allows us to approximate the number of people of group *g* living under the country *c*'s specific societal poverty line as follows:

$$povhc\_S\hat{P}L_{gct} = povrate\_S\hat{P}L_{gct} \times pop_{gct}, \qquad (2)$$

where $povhc\_S\hat{P}L_{gct}$ is the estimated number of individuals of group *g* below the SPL; and $povrate\_S\hat{P}L_{gct}$ is the estimated group-specific SPL poverty rate based on data on the overall SPL poverty rate and the poverty rate differential between vulnerable group *g* and rest of the population taken from secondary sources as follows:

$$povrate\_S\hat{P}L_{gct} = \frac{povrate\_SPL_{ct}}{sharepop_{gct} + \frac{1}{\hat{k}_{gct}}(1 - sharepop_{gct})}, \qquad (3)$$

where $\hat{k}_{gct} = \frac{povrate\_proxy_{gct}}{povrate\_proxy_{(-g)ct}}$ is the ratio between the available consistent poverty rates for at-risk group *g*, and the rest of the population (-*g*). In Annex 1A in S1 File, we present a detailed derivation and specification of Eq (3).

Thus, we use existing poverty estimates for specific vulnerable groups based on the international poverty line (IPL) that are relevant for the income level of each country. If a country is classified as an upper middle-income country, the poverty line assigned to each vulnerable group for which we have no information based on the SPL will be the World Bank IPL of USD 5.50 (2011 PPP) per person per day. Table 1 below summarizes all the approximations used for such vulnerable groups without specific societal poverty headcount estimates.

For example, we can estimate the number of indigenous people (IPs) living under the societal poverty line in Peru in 2017 as follows: First, based on [54], we estimate that there were 4.4 million IPs in Peru in 2017, roughly 14 percent of the total population. The estimated poverty rate based on the societal poverty line in Peru in 2017 was 26.64 percent [46], although

**Table 1. Proxy poverty lines used to extrapolate data gaps in the absence of group-specific SPLs.**

| Country, or economy, income level * | Proxy poverty lines |
|---|---|
| Low income | $1.90 IPL |
| Lower-middle income | $3.20 IPL |
| Upper-middle income (and high-income global south) | $5.50 IPL |
| High income | 50% median income or available national poverty line |

Source: Authors

Note: income level of the country or economy as defined by World Bank income level classification

according to [55]'s poverty estimates, 25.25 percent of IPs were poor in Peru in 2017 relative to a poverty rate of 22.9 percent among non-indigenous people. Based on this information and using Eq (3) as reference, we are able to estimate indigenous poverty rates based on the social

poverty line in Peru in 2017 as follows: $\left( \frac{0.2664}{0.14 + \frac{1}{\frac{25.25}{22.9}}(1-0.14)} \right) \times 100 = 28.96\%$. Finally, based on

Eq (2), we estimate that the number of IPs living under the societal poverty line in Peru in 2017 was: 0.2896×4.4 = 1.3 million. Annex 1A in S1 File provides a more extensive discussion of this process.

The third step identifies overlaps across estimates of populations at risk of exclusion in each country. To do so, we sequentially subtract those already accounted for from the total number of people at risk of being socially excluded estimated for each group in step 2. We assume independence between the different exclusion attributes when lacking information on the actual intersection. Additionally, we follow a consistent sequence across all countries: starting with poor children, then subtracting from the group of poor women the poor female children, then subtracting from the group of poor people with disabilities poor females (both adults and children) and poor male children and so on, as illustrated in Fig 2.

The *men* group is used as a residual group including those societal poor that do not belong to any of the preceding vulnerable groups. The total double-counted or overlapped at-risk populations do not change regardless of the sequence of this subtracting exercise. What does change if the subtracting sequence changes is the group-specific overlap with other populations (see S1 File). While our methodology allows to capture multiple overlaps across population groups (see S1 Table), it is only at the individual level that exclusion overlaps can be fully and precisely identified. This is possible when individual-level microdata captures information about her gender, location, age, access to markets, services, civic and political participation, disabilities, sexual orientation and gender identity, whether she has ever been displaced, and so forth.

Fig 2 illustrates the computation procedure. The first column shows the totals by at-risk group as computed in step 2. The second column enumerates the intersections between the associated total and the preceding groups using formal notation. In the third column we briefly describe those intersections. S1 File, section B, describes the step-by-step procedure to sequentially remove each overlap across populations at risk of exclusion.

The fourth step accounts for the number of those who are at risk of social exclusion because of their *circumstances*, that is, it accounts for the share of countries' populations that is victim to gender-based violence (GBV) and/or which is forcibly displaced. We compute the total number of women socially excluded due to their experience of violence in age group *a*, country *c* and year *t* as follows:

$$GBVhc_{(FE)act} = pop_{(FE)act} \times GBVrate_{(FE)act}, \tag{4}$$

where $pop_{(FE)act}$ refers to the total female population in age group $a$, country $c$, and year $t$; while $GBVrate_{(FE)act}$ stands for the incidence of gender-based violence in age group $a$, country $c$, and year $t$. For example, according to [56], in India, 22 percent of women aged 15 to 49 years were subjected to physical and/or sexual violence in the 12 months prior to 2016. India's estimated female population in that age category (15–49) was 340.8 million in 2016 [52]. Thus, based on Eq (4), we estimate that in 2016, there were 340.8×0.22 = 75 million women in India aged 15–49 years that were victims of physical and/or sexual violence. Regarding forcibly displaced populations, we take the incidence or headcount ratio for this group ($FDPhc_{ct}$) directly from official international statistics (see Section 4).

In a fifth step, to avoid double-counting among populations that are at risk of exclusion because of their circumstances, we proceed sequentially, starting from the total number of people in each country at risk of being excluded based on socioeconomic status (step 3), and add all women victims of gender-based violence minus those already accounted for because of their poverty status. We estimate the intersection between societal poverty and GBV based on data on the incidence of this group by income level or poverty status, following a procedure analogous to that used in step 2 to estimate the at-risk-group societal poverty headcounts. We note that that the incidence of GBV by income level or poverty status is not defined at the SPL in the available data. See Annex 1C in S1 File for a discussion of this procedure.

We then add all forcibly displaced persons not accounted for already by poverty status or for being a victim of GBV. The intersection between forcibly displaced populations and SPL ($povhc\_\hat{S}PL_{(FDP)ct}$) is estimated based on auxiliary data on the poverty differential between foreigners and nationals in a sample of countries, following the same procedure as in the second step discussed above. Finally, we estimate the intersection between non-poor forcibly displaced women and women subject to GBV, based on data on the incidence of GBV by poverty level and assuming that the incidence of GBV is constant across all age groups. See Annex 1C in S1 File for further details.

The last step adds together all estimated populations at risk of exclusion per country to obtain estimates of the global and regional populations at risk of social exclusion. To do so, we use estimates for those countries for which information on vulnerable populations is available and assign regional (or global) averages to those countries lacking data when such an extrapolation would be appropriate. Regarding Indigenous and forcibly displaced people, Afrodescendants and religious minorities, extrapolation is not conducted on countries failing to report the populations of these groups because two countries in the same region might have quite different population profiles based on their history or otherwise. Finally, for those cases where there is no information about the poverty incidence among at-risk populations, we extrapolate regional or global averages.

The advantages of the proposed methodology are manifold. Estimates are easy and quick to produce. Numbers and shares can be updated annually. Assumptions can be easily adjusted when better information emerges. It builds from a logical and consistent approach to avoid double counting that is replicable across countries and contexts. However, our methodology also runs up against important informational constraints. First, we do not observe social exclusion directly as opposed to risk factors of social exclusion, and the changing salience of those factors over time and across contexts. Second, we suspect a degree of measurement error involving double counting, as we use aggregated data from various sources. It is only at the micro level, using individual and household level data, that one can determine with greater precision whether a person is simultaneously affected by different forms of exclusion, which amplify the severity of exclusion, a key notion from a policy perspective. While double counting is mitigated in terms of the intersections between age, gender, poverty, disability, and

sexual orientation and gender identity, we have not been able to remove possible intersections between, for example, Afrodescendants and being a member of a religious minority. Such intersections are likely to be small, nevertheless, we note that in the absence of data that directly measures exclusion at the global scale, our methodology yields approximate—lower-bound—estimates of populations at risk of social exclusion. In the next section, we discuss the main data sources for empirical analysis.

## 4. Data

This section describes the data sources used to estimate the number and share of the global population at risk of exclusion. It also describes how the incidence of poverty is estimated for each of the populations vulnerable to exclusion following the methodology developed in the previous section. The sample includes 195 countries, comprising 99 percent of the world's population, for which data can be individually reported on these vulnerable populations. We use existing country-level data and impute regional averages to those countries lacking data for which extrapolation is appropriate, as discussed below. In many cases, we lack the exact data point for 2017, for which we impute the data from the closest available year and project forward or backward. In a few cases, including the LGBTI population group, we lack enough data to perform a regional extrapolation, so we use the global average instead.

S2 Table describes each of the data sources used for our vulnerable populations. To determine the overall vulnerable population we use population data from the World Bank [52] complemented with data from the United Nations [57]. We use population data disaggregated by gender and age for our sample of 195 countries. We define as children those of 0 to 17 years of age.

We use regional-level data from the World Health Organization and World Bank [58], which in turn publish data for 2004, to estimate the number of people with disabilities worldwide. These data are disaggregated by gender and age. Results refer to the prevalence of "severe" disability, defined as severity Classes VI and VII–the equivalent of being blind, or having Down Syndrome, quadriplegia, severe depression, or active psychosis. Estimates are also presented for "moderate and severe" disability, defined as severity Classes III and higher–the equivalent of having angina, arthritis, low vision, or alcohol dependence [58]: 296. Estimates based on the more stringent definition of disability are interpreted as lower bound estimates, whereas those based on the more extensive definition are interpreted as upper bound estimates. Because of the age grouping in the original sources, in some cases estimates for disabled children rely on data that refer to individuals ages 0 to 15. Aggregate data is provided for seven groupings of countries: the six WHO regions (limited to low- and middle-income countries), on the one hand, and high-income countries, on the other.

Regarding the LGBTI population, we use country-level data mostly from developed countries, as we did not find reliable estimates for developing countries (see S2 Table). Estimates refer mostly to adults, although some studies include adolescents (those over 15 or 16 years old). However, in those cases we assume estimates refer to the adult population to consistently derive adult-only estimates. All sources capture self-identified LGBTI people, but studies differ in the extent to which all members are covered from Lesbian, Gay, Bisexual, Transgender, Queer, Questioning, Intersex, Allies, Asexual and Pansexual (LGBTQ2+) in [59] to lesbian, gay and bisexual populations (LGB) in [60]. Only 14 countries have specific estimates. We extrapolate the average size of this group for countries with no data. Estimates based on all the available data, irrespective of the actual definition of the group in the corresponding data source, are interpreted as lower bound estimates, as they are lower than alternative estimates based on the extrapolation of [61]'s estimation of the LGBTI population in the United States, which are reported as upper bound estimates.

Data on Indigenous peoples (IPs) by country are drawn from [53, 54, 62, 63], who compile data from several sources. We use the lower estimate reported in any of the available data sources as a lower bound estimate for the total number of indigenous peoples in a country and the larger estimate found as an upper bound estimate. In total, we have data for 59 countries. There is no imputation for this category, as it is not possible to make assumptions about the share of IPs in a given country from others, even those in the same region.

Data on the Afrodescendant population by country are drawn from the following sources: [64], which includes data for 30 countries; [65], which includes data for 11 Latin American countries; [59] for Canada; [66] for the UK; [67] for the US; [68] for France; [69] for Italy; [70] for Spain; [71] for Saudi Arabia; and [72] for Yemen. We have data for a total of 44 countries. As in the case of IPs, there is no country imputation for this category.

Data on religious minority populations are drawn from [73]: 8, which covers 233 countries and territories. Categories include Buddhists, Christians, Folk Religions (including African traditional religions, Chinese folk religions, Native American religions, and Australian Aboriginal religions), Hindus, Jews, Muslims, Other Religions (an umbrella category that includes Baha'is, Jains, Sikhs, Taoists, and many smaller faiths, the largest of this group being Sikhs), and Unaffiliated. We compute the share of population belonging to a religious minority as the total share of those religious affiliations that account separately for less than the 25 percent of the country's population. This allows us to include Muslims as minorities in India and Israel, as they represent 15.4 and 20.1 percent of these countries' populations, respectively. At the same time, this threshold avoids including large religious groups as minorities, such as Christians in Nigeria, who represent 46.9 percent of the population, but are the dominant religion in southern regions.

Regarding female victims of gender-based violence, we estimate the proportion of women subjected to physical and/or sexual violence in the last 12 months provided by [74]. We estimate the number of GBV victims who are children (15–17 years old) and those who are adult (18–49 years old) assuming that the incidence is constant across age groups. The number for the adult population represents a lower bound estimate that is complemented with an upper bound estimate that extrapolates the incidence of GBV to the whole adult female group. We have data for 103 countries. We extrapolate the regional average incidence of GBV to those countries lacking data using regional and income levels.

Data on forcibly displaced populations (FDP) is drawn from [75]. For each asylum country and year, we compute the total number of FDPs as the sum of refugees under the United Nations High Commissioner for Refugees (UNHCR)'s mandate; asylum-seekers; IDPs of concern to UNHCR; Venezuelans displaced abroad; stateless persons; and others of concern over all countries of origin.

After each vulnerable population at the country, regional, and global level is estimated, we compute the share of those populations living in poverty. In order to do so, we start from the country's overall societal poverty line reported by [46], available for 164 countries of the sample and covering 97 percent of the world's population. We extrapolate the regional average incidence of societal poverty incidence to those countries lacking data. We then estimate the share of the societal poor who belong to each at risk of exclusion group based on available country-level data on the incidence of poverty among each specific group.

We rely on additional sources to estimate the shares of distinct groups in each country's societal poverty headcount (see S3 Table). By age and sex, and disability, LGBTI, Afrodescendant, religious minority, and forcibly displaced status, the available data allows us to determine consistent poverty rates at the corresponding poverty line for the relevant group and for the overall population. The regional or global averages of these poverty rates are extrapolated to those countries lacking data, which are used to estimate the incidence of societal poverty

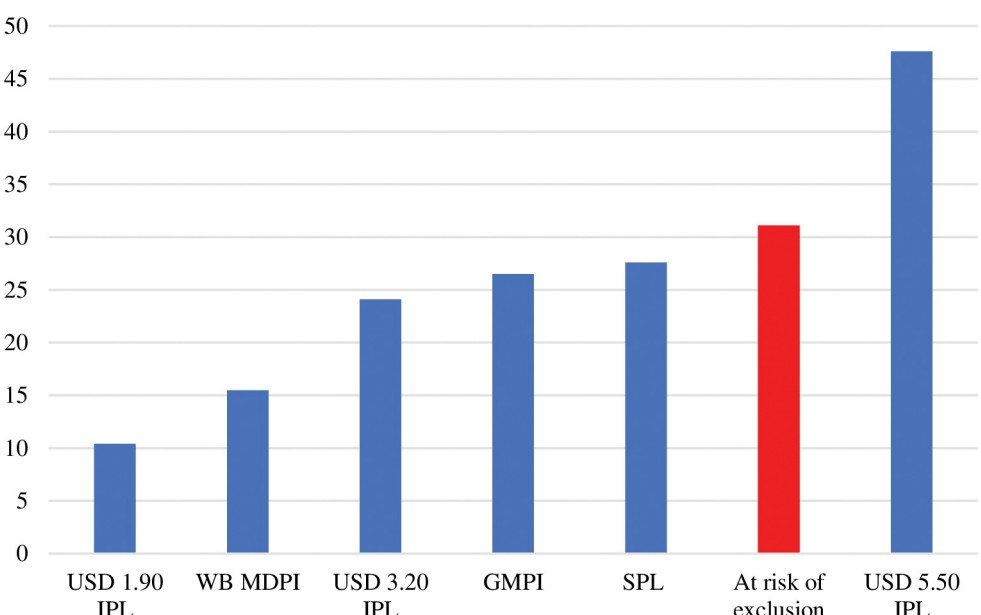

**Fig 3. Global rates of poverty and exclusion, %, 2017.** Source: Authors' estimates. Note: IPL, international poverty line; MDPI, multidimensional poverty index; GMPI, global multidimensional poverty index; SPL, societal poverty line.

among the relevant group as explained in the previous section. There are slightly different procedures for forcibly displaced and indigenous peoples, and victims of GBV, explained below.

Regarding age, we use data from [1, 76]. There are a total of 149 countries with poverty data available by age group. Similarly, we use data from [60, 77] which provide poverty data disaggregated by gender. We have poverty data by gender for 156 countries. For each country we use poverty data at the income threshold that is closer to its societal poverty line. We use a regional average to estimate societal poverty by age and gender for countries lacking data.

For the incidence of poverty among people with disabilities, we rely on several sources, each of which uses a different methodology for estimating poverty. [78] use the US Census Bureau's multi poverty thresholds measure. [3] has data for 36 European countries, using the at-risk-of-poverty measure. [79] have data for 15 developing countries, using [80]. We use a regional average to estimate societal poverty among people with disabilities for those countries lacking data.

As far as we know, only two studies estimate the incidence of poverty for LGBTI people as it compares with non-LGBTI people, using 50 percent of the median income in the case of the UK [81] and the national poverty line in the case of the US [82]. We use the average of these two poverty rates to estimate societal poverty among LGBTI people in countries lacking data.

To estimate the incidence of poverty among indigenous peoples, we use two sources [55, 83], Fig 3.11 in pp. 95–97, which presents poverty headcount using the $1.90, $3.20, and $5.50 a day poverty lines by indigenous peoples' status and sex for Africa, Asia and the Pacific, Latin America and the Caribbean, North America, low-income countries, lower middle-income countries, upper middle-income countries, and high-income countries. We impute the corresponding regional poverty rates to individual countries, except for the case of Australia, Japan, and New Zealand which are given high-income-country regional values instead of Asia-Pacific values. Moreover, we impute the corresponding income class averages to European countries with IPs given that there are not regional data available in their case. Finally, we use [55]'s country-level data for 11 Latin American countries, which provides poverty rates by ethnicity/race and gender groups at two different poverty lines: poverty and extreme poverty. As previously, for each

country we use the poverty line that is closer to its societal poverty line. We use a regional average to estimate societal poverty for these groups in countries lacking data.

Regarding the incidence of poverty among Afrodescendants, we use the following sources. First, [55], second, [78], which provide poverty rates for Black Americans and the population at large in the US. Third, [72], which provides a description of the socioeconomic status of AD people in Yemen. We use a world average of these poverty rates to estimate societal poverty among these groups within countries lacking data.

Poverty data concerning religious minorities are available for India [84] and the UK [85]. We transform the poverty rates provided in these sources using refined religious population data from [86] for India, and [66, 87] for the UK, to get poverty rates consistent with [73] religious categories. We extrapolate the Indian poverty rates to low- and middle-income countries, and the UK ones to high-income countries. For the intersection between GBV and poverty, we collect information on the incidence of GBV by income level/poverty status from several sources, as shown in S3 Table. As we have data on the probability of poverty across GBV victims for at least one country from every world region, we impute the average regional GBV poverty likelihood to those countries that lack such data in the same region.

Regarding the incidence of poverty within the group of forcibly displaced people, we rely on data for the EU provided by [3] on people at risk of poverty or social exclusion by broad group of citizenship and gender for the population aged 18 and over, and the at risk of poverty rate for children by citizenship of their parents for the population aged 0 to 17 years. We use this European data to consistently estimate the poverty rates for foreigners and nationals by age and gender group and for the overall population. The average of these poverty rates is extrapolated to those countries lacking data, allowing us to estimate the incidence of societal poverty among forcibly displaced people.

## 5. Results

Results indicate that between 2.33 and 2.43 billion people worldwide are at risk of social exclusion (Table 2). This represents between 31.1 and 32.4 percent of the global population in 2017. The difference between the lower and upper bound estimates, that is, 102 million people, is explained by the higher or lower estimates of GBV victims.

The main contributors to social exclusion—after double-counting is deducted—relate to women, children, and poor men categories. They represent between 85 and 90 percent of the at risk of exclusion populations in the upper and lower bound estimates, respectively. By contrast, LGBTI, Indigenous, Afrodescendant, and forcibly displaced populations represent about 100 million or around 4 percent of the at-risk population globally.

Regional differences are wide with respect to both absolute numbers and shares within regional populations (see Table 3). At risk population totals are highest in South Asia at between 622 million and 644 million, followed closely by East Asia and Pacific at 609 million (659 million as upper bound estimate), and sub-Saharan Africa at 552 million (558 million). These three regions amount for 76 percent of the global population at risk of exclusion and 68 percent of the global population.

Unsurprisingly, India and China—the most populous countries in the world—concentrate most of the at-risk populations in their respective regions. India reports approximately 470.9 million people of at risk of exclusion, while China is home to 369.2 million (using lower-bound estimates). These numbers represent 75 and 60 percent of the entire population at risk of exclusion in South Asia and East Asia and Pacific, respectively, and 36 percent of all people at risk of exclusion worldwide. These findings do not change when considering upper-bound estimates (Table 3).

**Table 2. Global vulnerable populations by group at risk of social exclusion and social exclusion dimension: Lower and upper bounds, 2017.**

|  | Lower bound | | Upper bound | |
|---|---|---|---|---|
|  | N (millions) | % Total | N (millions) | % Total |
| **Children** |  |  |  |  |
| *Population* | 2,317.3 | 30.9 | 2,317.3 | 30.9 |
| Poor | 767.3 | 37.0 | 767.3 | 37.0 |
| *Socially excluded not double counted* | **767.3** |  | **767.3** |  |
| **Women** |  |  |  |  |
| *Population* | 3,722.2 | 49.6 | 3,722.2 | 49.6 |
| Poor | 1,038.4 | 50.0 | 1,038.4 | 50.0 |
| (Intersection: CH) | (383.6) |  | (383.6) |  |
| *Socially excluded not double counted* | **654.8** |  | **654.8** |  |
| **People with disabilities** |  |  |  |  |
| *Population* | 225.9 | 3.0 | 1,182.4 | 15.8 |
| Poor | 74.4 | 3.6 | 385.0 | 18.5 |
| (Intersection: CH and FE) | (40.8) |  | (215.6) |  |
| *Socially excluded not double counted* | **33.6** |  | **169.4** |  |
| **LGBTI people** |  |  |  |  |
| *Population* | 132.7 | 1.8 | 228.3 | 3.0 |
| Poor | 36.0 | 1.7 | 61.7 | 3.0 |
| (Intersection: FE and PWD) | (24.0) |  | (47.6) |  |
| *Socially excluded not double counted* | **12.0** |  | **14.1** |  |
| **Indigenous people** |  |  |  |  |
| *Population* | 327.0 | 4.4 | 357.1 | 4.8 |
| Poor | 124.9 | 6.0 | 134.7 | 6.5 |
| (Intersection: CH, FE, PWD & LGBTI) | (86.5) |  | (103.2) |  |
| *Socially excluded not double counted* | **38.4** |  | **31.5** |  |
| **Afrodescendants** |  |  |  |  |
| *Population* | 188.1 | 2.5 | 188.1 | 2.5 |
| Poor | 59.5 | 2.9 | 59.5 | 2.9 |
| (Intersection: CH, FE, PWD & LGBTI) | (41.8) |  | (46.2) |  |
| *Socially excluded not double counted* | **17.7** |  | **13.3** |  |
| **Religious minorities** |  |  |  |  |
| *Population* | 1,527.1 | 20.3 | 1,527.1 | 20.3 |
| Poor | 432.2 | 20.8 | 432.2 | 20.8 |
| (Intersection: CH, FE, PWD, LGBTI & IP) | (328.4) |  | (352.8) |  |
| *Socially excluded not double counted* | **103.8** |  | **79.4** |  |
| **Men** |  |  |  |  |
| *Population* | 3,784.5 | 50.4 | 3,784.5 | 50.4 |
| Poor | 1,037.1 | 50.0 | 1,037.1 | 50.0 |
| (Intersect.: CH, PWD, LGBTI, IP, AD & RELI) | (589.1) |  | (691.5) |  |
| *Socially excluded not double counted* | **448** |  | **345.6** |  |
| **Victims of gender-based violence** |  |  |  |  |
| *Victims of gender-based violence* | 334.7 | 4.5 | 481.8 | 6.4 |
| (Intersection: SPL poor) | (113.5) |  | (158.0) |  |
| *Socially excluded not double counted* | **221.3** |  | **323.8** |  |
| **Forcibly displaced people** |  |  |  |  |
| *Forcibly displaced people* | 67.6 | 0.9 | 67.6 | 0.9 |
| (Intersection: SPL poor & victims GBV) | (32.1) |  | (32.6) |  |

*(Continued)*

**Table 2.** (Continued)

| | Lower bound | | Upper bound | |
|---|---|---|---|---|
| | N (millions) | % Total | N (millions) | % Total |
| *Socially excluded not double counted* | 35.5 | | 35.5 | |
| **TOTAL** | | | | |
| *Population* | 7,506.7 | 100.0 | 7,506.7 | 100.0 |
| *Socially excluded* | 2,332.2 | 31.1[a] | 2,434.3 | 32.4[a] |

Source: Authors' estimates.

Notes: ([a]) % World total population. See section 3 for further details on intersections. The lower bound estimates are based on the following assumptions: (1) victims of GBV are restricted to women aged 15 to 49; (2) the population of PWD is based on the most restrictive definition of disability (severe disability); (3) the population of LGBTI people is based on all available data; (4) the population of IPs is based on the lower estimate available in our sources for each country. On the other hand, the upper bound estimates are based on the following assumptions: (1) the prevalence of GBV is extrapolated to the whole adult female group; (2) the population of PWD is based on the least restrictive definition of disability (at least moderate disability); (3) the size of the LBGTI group in the US (4.4%) is extrapolated to the rest of countries; (4) the population of IPs is based on the larger estimate available in our sources for each country. See section 4 for further details.

The share of the at-risk population in sub-Saharan Africa is the largest worldwide. More than half of the total population (roughly 53 percent) is found to be at risk of exclusion based on both lower and upper bound estimates (Table 3). Its rate doubles that of East Asia and Pacific (26 percent), and well exceeds that of MENA (28 percent), South Asia (34 percent), Latin America (29 percent), North America (20 percent) and ECA (17 percent). It is worth noting that while North America (comprising US and Canada) has the smallest population at risk of exclusion of any region, the share of its population at risk of exclusion exceeds that of Europe and Central Asia.

We also observe a significant variation in the contribution of factors of exclusion to absolute and relative estimates of at-risk populations *within* regions. For instance, while children represent between 22 and 24 percent of total at risk populations in North America, Europe and Central Asia, East Asia and Pacific, based on lower and upper bound estimates respectively, those shares increase to 30–34 percent in Latin America and the Caribbean, and South Asia, and then to 38–47 percent in sub-Saharan Africa and the MENA region. Women come as the second most vulnerable group to social exclusion after children in Latin America, South Asia, the MENA region and sub-Saharan Africa, but become the most vulnerable group in East Asia and the Pacific, Europe and Central Asia, and North America (see S4 to S10 Tables).

Estimates also indicate that the prevalence of gender-based violence as a factor of exclusion is present across all world regions but particularly acute in South Asia and East Asia and Pacific where approximately 11–14 percent and 14–20 percent of total at risk populations, respectively, belong to this group, based on lower and upper bound estimates. Furthermore, the largest shares of indigenous populations at risk of exclusion are observed in Latin America and the Caribbean, South Asia and East Asia and Pacific, with a contribution to total at-risk populations in the order of 2 and 3 percent, while Afrodescendants contribute to approximately 5 to 7 percent of at-risk populations in Latin America and the Caribbean and North America. Religious minorities are vulnerable to social exclusion in South Asia, East Asia and Pacific, and Europe and Central Asia, where these groups contribute to 4 to 7 percent of total at-risk populations.

Estimates using the World Bank's country classification by income level show that lower-income economies report the highest incidence of people at risk of exclusion, with 50.9 percent of their population (lower bound; 51.7 percent upper bound—see Table 4). But with a small share of worldwide population, lower-income economies are only home to 6 percent of the

**Table 3. World vulnerable population by group at risk of social exclusion: Lower and upper bounds, by region 2017.**

| | Lower bound | | | Upper bound | | |
|---|---|---|---|---|---|---|
| | N (millions) | % Population | % Global Total | N (millions) | % Population | % Global Total |
| **Latin America** | | | | | | |
| *Population* | 634.2 | 100.0 | 8.4 | 634.2 | 100.0 | 8.4 |
| *Social exclusion* | **187.9** | **29.6** | **8.1** | **194.3** | **30.6** | **8.0** |
| **South Asia** | | | | | | |
| *Population* | 1,792.9 | 100.0 | 23.9 | 1,792.9 | 100.0 | 23.9 |
| *Social exclusion* | **622.1** | **34.7** | **26.7** | **644.2** | **35.9** | **26.5** |
| Without India | | | | | | |
| *Population* | 454.2 | 100.0 | 6.1 | 454.2 | 100.0 | 6.1 |
| *Social exclusion* | **151.2** | **33.3** | **6.5** | **156.7** | **34.5** | **6.4** |
| **East Asia Pacific** | | | | | | |
| *Population* | 2,314.0 | 100.0 | 30.8 | 2,314.0 | 100.0 | 30.8 |
| *Social exclusion* | **609.5** | **26.3** | **26.1** | **659.7** | **28.5** | **27.1** |
| Without China | | | | | | |
| *Population* | 927.6 | 100.0 | 12.4 | 927.6 | 100.0 | 12.4 |
| *Social exclusion* | **240.3** | **25.9** | **10.3** | **258.79** | **27.9** | **10.6** |
| **Europe and Central Asia** | | | | | | |
| *Population* | 912.5 | 100.0 | 12.2 | 912.5 | 100.0 | 12.2 |
| *Social exclusion* | **161.8** | **17.7** | **6.9** | **171.5** | **18.8** | **7.0** |
| **MENA** | | | | | | |
| *Population* | 441.3 | 100.0 | 5.9 | 441.3 | 100.0 | 5.9 |
| *Social exclusion* | **124.3** | **28.2** | **5.3** | **127.3** | **28.8** | **5.2** |
| **North America** | | | | | | |
| *Population* | 361.7 | 100.0 | 4.8 | 361.7 | 100.0 | 4.8 |
| *Social exclusion* | **74.3** | **20.5** | **3.2** | **79.0** | **21.8** | **3.2** |
| **Sub-Sahara Africa** | | | | | | |
| *Population* | 1,050.2 | 100.0 | 14.0 | 1050.2 | 100.0 | 14.0 |
| *Social exclusion* | **552.3** | **52.6** | **23.7** | **558.3** | **53.2** | **22.9** |
| **World total social exclusion** | **2332.2** | **31.1** | **100.0** | **2,434.3** | **32.4** | **100.0** |

Source: Authors' estimates.

Notes: See S4 to S10 Tables for definitions of lower and upper bound estimates by world regions.

worldwide at risk of exclusion population. Middle-income economies, by contrast, represent two thirds of both the global population and the population at risk of exclusion. The population at risk of exclusion in lower-middle-income economies combined is the most numerous and has a largest share across regions. High-income economies represent 16 percent of the global population but less than 10 percent of the worldwide population at risk of exclusion. As expected, people in fragile, conflict-affected, and violent contexts are highly likely to be at risk of exclusion. About 48 percent of the population in such contexts are categorized as such, a reflection of their geographical composition (see Table 4). Again, all these findings hold for upper-bound estimates.

These global and regional estimates confirm that the global population at risk of exclusion exceeds most existing estimates of monetary and multidimensional poverty. With a lower bound estimate of 31.1 percent of the global population in 2017, there were three times more people at risk of exclusion that than those living under the absolute monetary poverty line of USD 1.90 per person per day (2011 PPP)—estimated by [2] to be around 10.4 percent. They

**Table 4. Global estimates of social exclusion by World Bank income category and fragile and conflict-affected situations for 2017: Lower and upper bounds.**

| | Lower bound | | | Upper bound | | |
|---|---|---|---|---|---|---|
| | N (millions) | % Population | % Global Total | N (millions) | % Population | % Global Total |
| **High-income economies** | | | | | | |
| *Population* | 1,247.7 | 100.0 | 16.6 | 1,247.7 | 100.0 | 16.6 |
| *Social exclusion* | **225.8** | 18.1 | **9.7** | **246.2** | 19.7 | **10.1** |
| **Upper-middle-income econ.** | | | | | | |
| *Population* | 2,472.5 | 100.0 | 32.9 | 2,472.5 | 100.0 | 32.9 |
| *Social exclusion* | **656.0** | 26.5 | **28.1** | **700.3** | 28.3 | **28.8** |
| **Lower-middle-income econ.** | | | | | | |
| *Population* | 2,645.6 | 100.0 | 35.2 | 2,645.6 | 100.0 | 35.2 |
| *Social exclusion* | **893.1** | 33.8 | **38.3** | **923.6** | 34.9 | **37.9** |
| **Low-income economies** | | | | | | |
| *Population* | 273.8 | 100.0 | 3.6 | 273.8 | 100.0 | 3.6 |
| *Social exclusion* | **139.5** | 50.9 | **6.0** | **141.5** | 51.7 | **5.8** |
| **Fragile & conflict-affected Situations** | | | | | | |
| *Population* | 867.1 | 100 | 11.6 | 867.1 | 100.0 | 11.6 |
| *Social exclusion* | **417.8** | 48.2 | **17.9** | **422.6** | 48.7 | **17.4** |
| **World total social exclusion** | **2,332.2** | **31.1** | **100.0** | **2,434.3** | **32.4** | **100.0** |

Source: Authors' estimates.

Notes: See S4 to S10 Tables for definitions of lower and upper bound estimates by world regions.

also more than doubled the share of the multidimensionally poor, at 15.5 percent [2]. Those vulnerable to exclusion also exceeded global estimated numbers of those living in monetary poverty at USD 3.20 per person per day (24.1 percent), OPHI's estimates of the multidimensionally poor using the Global Multidimensional Index (26.5 percent), and estimates using societal poverty lines (27.6 percent). Only global monetary poverty estimates using the USD 5.50 international poverty line, at 47.6 percent, exceed the share of the population at risk of exclusion (see Fig 3).

# 6. Conclusions

This paper contributes to the literature on social exclusion in three import domains: First, on a conceptual level, we develop a framework that conceptualizes social exclusion holistically, emphasizing its relative, agential, multidimensional, and dynamic nature. This framework encapsulates identity, circumstances, and the socioeconomic conditions as key dimensions of exclusion among vulnerable populations.

Second, methodologically, we employ a macro-counting approach to estimate the proportions of individuals within vulnerable populations that are at risk of exclusion. This methodology addresses information gaps by imputing values based on regional or global peer averages and incorporates a protocol to control for double counting across categories of vulnerable populations. Due to a lack of sufficient comparable microdata, we cannot observe directly whether individuals are unable to access services, markets, and spaces, live under dignified conditions, or exercise meaningful decision-making. Instead, our macro-counting approach helps fill this data gap by providing upper- and lower-bound estimates of populations at risk of social exclusion.

Third, empirically, we present, to the best of our knowledge, the first estimates of populations at risk of social exclusion on a global and regional scale. The results indicate that between

2.33 and 2.43 billion people, or between 31.1 and 32.4 percent of the global population, are at risk of exclusion based on identity, circumstances, and/or socioeconomic considerations. This number is notably higher than global estimates of monetary poverty at the USD 1.90 (2011 PPP) international poverty lines and the World Bank's multidimensional poverty headcounts. It is also moderately larger than the poverty rates at the USD 3.20 (2011 PPP) international poverty line, the Global Multidimensional Poverty headcounts, and the global incidence of societal poverty as defined by the World Bank.

Vulnerability to exclusion is most widespread in absolute terms in the South Asia and East Asia and Pacific regions. This is not surprising given the large populations of several countries in that world region. Incidence is highest in sub-Saharan Africa, where over 52 percent of people are at risk of exclusion. This rate is three times that observed in Europe and Central Asia. The share of at-risk populations in fragile, conflict-affected and violent contexts is close to 50 percent, driven by high rates in sub-Saharan Africa (and to a lesser extent South Asia and East Asia Pacific). As expected, fewer people are at risk of exclusion in high-income economies (less than 10 percent of their population) while more people are vulnerable in low-income economies (about 48 percent). Middle income economies, the most populous worldwide and with large shares of exclusion—bordering 40 percent in the case of lower-middle-income economies—concentrate two thirds of those at risk of exclusion worldwide.

Our macro counting method has several limitations. First, data gaps exist but are more pronounced across fragile and poor countries. This is only partially compensated by the fact that these countries represent a small share of the global population. Second, even though double counting is dealt with methodically, it cannot be fully eliminated. To do so would require precise microdata that could identify each exclusion factor at the individual level (and specific thresholds to define exclusion sensitive to context). The closest data source to do this is a census. However, censuses are neither frequently collected, nor do they typically allow to identify certain vulnerable groups such as LGBTI people, forcibly displaced populations, or victims of GBV. Opportunities to improve such microdata to eliminate risks of double counting would require technical innovations that avoid jeopardizing the privacy of such groups. Third, our analysis provides a static snapshot of exclusion at a point in time, but does not capture the fact that the roots, drivers, and agents of exclusion may well change over time and space.

Notwithstanding these data limitations, global estimates of populations at risk of social exclusion provide three relevant policy messages. First, policies targeted at alleviating or ending most extreme forms of poverty might overlook the non- poor who are nevertheless at risk of exclusion. Globally, this gap is substantial in magnitude: about a fifth of the world's population, or almost 1.5 billion people.

Broadening access to labor, financial, and land markets, as well as improving the coverage and quality of basic service provision, may effectively reduce both poverty and exclusion when widely applied within a society. However, poverty-targeted interventions, such as cash or in-kind transfers that successfully increase consumption and smooth its volatility among the extremely poor, may be less effective in reducing exclusion stemming from racial or gender discrimination.

Second, effectively addressing exclusion necessitates multiple interventions tailored to distinct groups and sustained over time. Exclusion factors related to ethnicity, skin color, or religious affiliation can be acute in certain contexts but not in others. Drivers of social exclusion, such as discriminatory laws, social norms, weak institutions, and recurrent crises, may be common to all excluded groups. However, exclusion due to gender-based violence (GBV) or forced displacement requires a set of interventions that might not be effective in combating exclusion arising from long-term unemployment or lack of access to health or financial services.

Third, the complexities in defining and measuring exclusion do not render it unmanageable. Nonetheless, effective addressing requires precise estimates and agreed-upon

methodologies derived from available sources that can be frequently and easily monitored. Various similar methodological challenges have been successfully tackled in the past, such as global food price monitoring and acute food insecurity warning systems, internationally agreed-upon systems of national accounts, definitions and measurements of decent work, and international statistics on crime and justice. These experiences offer valuable insights into the importance of concerted action, achieving technical agreements, operationalizing monitoring, and utilizing data for effective policymaking.

Yet, a macro approach, no matter how improved, cannot substitute for a comprehensive micro approach where individual data revealing identity, circumstances, and socioeconomic conditions can be compared. This necessitates significant efforts in harmonizing household surveys globally, expanding the number and nature of questions typically collected in censuses, and improving sampling frames to capture representative samples of vulnerable, marginalized, and often invisible populations.

Efforts towards more inclusive and harmonized data are already underway. Examples include the Integrated Public Use Microdata Series managed by the Minnesota Population Center, the development of disability statistics suitable for censuses and national surveys, the Global Monitoring Database by the World Bank for producing comparable poverty statistics worldwide, and the Inclusive Data Charter auspiced by the UN. However, more and better-coordinated efforts are still needed before we can develop a micro approach to measure social exclusion at a global level.

## Supporting information

**S1 File. Supplementary information on methodology.**
(DOCX)

**S1 Table. Number of socially excluded populations.**
(DOCX)

**S2 Table. Data sources.**
(DOCX)

**S3 Table. Poverty studies on vulnerable populations.**
(DOCX)

**S4 Table. Populations at risk of social exclusion in LAC.**
(DOCX)

**S5 Table. Populations at risk of social exclusion in SA.**
(DOCX)

**S6 Table. Populations at risk of social exclusion in EAP.**
(DOCX)

**S7 Table. Populations at risk of social exclusion in ECA.**
(DOCX)

**S8 Table. Populations at risk of social exclusion in MENA.**
(DOCX)

**S9 Table. Populations at risk of social exclusion in NA.**
(DOCX)

**S10 Table. Populations at risk of social exclusion in SSA.**
(DOCX)

**S1 Data.**
(DO)

**S2 Data.**
(DTA)

## Acknowledgments

The authors are grateful to Maitreyi Das, Louise Cord, Richard Damania, Ambar Narayan, Mario Negre, Hugo Ñopo, Michael Woolcock, the Editor and two anonymous referees for useful comments and suggestions on previous versions of this paper. All errors are solely the responsibility of the authors.

## Author Contributions

**Conceptualization:** Jose Cuesta, Miguel Niño-Zarazúa.

**Data curation:** Borja López-Noval, Miguel Niño-Zarazúa.

**Formal analysis:** Jose Cuesta, Borja López-Noval, Miguel Niño-Zarazúa.

**Investigation:** Jose Cuesta.

**Methodology:** Borja López-Noval, Miguel Niño-Zarazúa.

**Writing – original draft:** Jose Cuesta, Borja López-Noval, Miguel Niño-Zarazúa.

**Writing – review & editing:** Jose Cuesta, Borja López-Noval, Miguel Niño-Zarazúa.

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
