## [Decision Letter · Decision Letter 0]

31 Aug 2023

PONE-D-23-14600Social Exclusion: Concepts, Measurement, and a Global EstimatePLOS ONE

Dear Dr. Nino-Zarazua,

Thank you for submitting your manuscript to PLOS ONE. After careful consideration, we feel that it has merit but does not fully meet PLOS ONE’s publication criteria as it currently stands. Therefore, we invite you to submit a revised version of the manuscript that addresses the points raised during the review process.

We look forward to receiving your revised manuscript.

Kind regards,

Vinay Kandpal

Academic Editor

PLOS ONE

Journal Requirements:

Additional Editor Comments:

Some of the inputs which will enhance the quality of paper:

Abstract needs to highlight research gap, purpose, findings and implications

Introduction needs to be motivated further highlighting the research gaps with relevant recent literature. Purpose of the study and benefit for the global audience should be clear.

Literature Review should be extensive with updated literature. Less use of older papers/literature. Theme wise presentation and sequencing should be checked.

Underpinning theory needs to be discussed.

Proof reading required.

Contribution of paper needs to be discussed.

Research methods could be better explained with relevant citations of literature highlighting the use of particular method.

Reviewers' comments:

Reviewer's Responses to Questions

**Comments to the Author**

1. Is the manuscript technically sound, and do the data support the conclusions?

Reviewer #1: Yes

Reviewer #2: Yes

2. Has the statistical analysis been performed appropriately and rigorously? 

Reviewer #1: Yes

Reviewer #2: Yes

3. Have the authors made all data underlying the findings in their manuscript fully available?

Reviewer #1: Yes

Reviewer #2: Yes

4. Is the manuscript presented in an intelligible fashion and written in standard English?

Reviewer #1: Yes

Reviewer #2: Yes

5. Review Comments to the Author

Reviewer #1: Observations:

The study highlights the concept of social exclusion in individual, regional, and global level, and also highlights the suggestions to addressing the problem of social exclusion.

The study estimates the share and number of population at risk of social exclusion in regional and global level.

The study finds that 2.33 and 2.43 billion people are at risk of exclusion worldwide, highest in sub-Saharan Africa, where over 52 percent of people are at risk of exclusion.

The study also finds the share of at-risk populations in fragile, conflict-affected and violent context worldwide is close to 50 percent, driven by high rates in sub-Saharan Africa (and to a lesser extent South Asia and East Asia Pacific). The paper is well written.

Minor Comments:

How did you define the social exclusion group? Please justify. In figure 1, and 2 provide source. In methodology section justify your formulas, which you used along with description. Description of variables and data table should be included.

Reviewer #2: The article is interesting and relevant in terms of the methodology developed. There are 3-4 following sections of the article which need strengthening.

1. Title: the current title needs to communicate the crux of the research and therefore the authors are suggested to rethink the title. The title should contain the purpose for this kind of study is useful, for instance, supporting global policy decisions on social inclusion in development.

2. Abstract : the abstract needs to be strengthened with inclusion of the key implications of the research in the concluding sentences.

3. Conceptualizing social exclusion: In this section the authors can use a timeline to show how how the concept of social exclusion has evolved over time.

4. Methodology: the authors should use a numerical example to explain the equations for comprehension of the readers.

5. Conclusion: the key points emerging from the analysis and how they relate to the wider context needs to be strengthened in this section.

Other minor suggestions: Grammatical errors, and the errors in the references need to be corrected.

6. PLOS authors have the option to publish the peer review history of their article (what does this mean?). If published, this will include your full peer review and any attached files.

Reviewer #1: **Yes: **Dr. Narendra N. Dalei

Reviewer #2: **Yes: **Suparana Katyaini

---

## [Author Response · Author response to Decision Letter 0]

18 Oct 2023

Responses to Editor 

A professional proof reader and editor has looked at the revised version and formatted the manuscript according to PLOS style requirements.

We note that the grant information you provided in the ‘Funding Information’ and ‘Financial Disclosure’ sections do not match. When you resubmit, please ensure that you provide the correct grant numbers for the awards you received for your study in the ‘Funding Information’ section. a) Please clarify the sources of funding (financial or material support) for your study. List the grants or organizations that supported your study, including funding received from your institution. b) State what role the funders took in the study. If the funders had no role in your study, please state: “The funders had no role in study design, data collection and analysis, decision to publish, or preparation of the manuscript.” c) If any authors received a salary from any of your funders, please state which authors and which funders. d) If you did not receive any funding for this study, please state: “The authors received no specific funding for this work.” Please include your amended statements within your cover letter; we will change the online submission form on your behalf.

As indicated, we have included in the cover letter a paragraph stating that “this study was supported by the World Bank. The funder had no role in study design, data collection and analysis, decision to publish, or preparation of the manuscript. One author, Jose Cuesta, receives a salary from the funder.”

In your Data Availability statement, you have not specified where the minimal data set underlying the results described in your manuscript can be found. PLOS defines a study's minimal data set as the underlying data used to reach the conclusions drawn in the manuscript and any additional data required to replicate the reported study findings in their entirety. All PLOS journals require that the minimal data set be made fully available. For more information about our data policy, please see http://journals.plos.org/plosone/s/data-availability. Upon re-submitting your revised manuscript, please upload your study’s minimal underlying data set as either Supporting Information files or to a stable, public repository and include the relevant URLs, DOIs, or accession numbers within your revised cover letter. For a list of acceptable repositories, please see http://journals.plos.org/plosone/s/data-availability#loc-recommended-repositories. Any potentially identifying patient information must be fully anonymized. We will update your Data Availability statement to reflect the information you provide in your cover letter. 

Regarding the Data Availability statement, we have submitted two Stata files. The first file contains the data used to estimate the populations at risk of social exclusion, and the second one is the Stata do. file that constructs the data set drawing from the different data sources cited in the paper, and estimates the populations at risks of exclusion based on the methodology and procedures described in the paper. 

We note that you have stated that you will provide repository information for your data at acceptance. Should your manuscript be accepted for publication, we will hold it until you provide the relevant accession numbers or DOIs necessary to access your data. If you wish to make changes to your Data Availability statement, please describe these changes in your cover letter and we will update your Data Availability statement to reflect the information you provide.

We have stated on the cover letter that since we are submitting the dataset in Stata format, we no longer plan to provide a repository information for such information, and therefore, wish to amend the previous Data Availability statement.

Please include captions for your Supporting Information files at the end of your manuscript, and update any in-text citations to match accordingly. Please see our Supporting Information guidelines for more information: http://journals.plos.org/plosone/s/supporting-information.

We have included captions for the Supporting Information files at the end of the manuscript, and submitted the actual materials as a separate file.

Additional comments by the Editor

Abstract needs to highlight research gap, purpose, findings and implications.

Thanks for the comment. The revised abstract now highlight the research gaps and underscores the key contributions of the paper. The revised paper also emphasize the purpose of this paper, which is to improve the knowledge base about the global scale of social exclusion and its most likely sources. The abstract also calls the attention of the reader to our findings, namely, the number and share of people at risk of exclusion, and rates/shares by world regions and types of countries according to fragility and income levels. The revised abstract also enumerates the contributions of the paper in a more explicit manner. Three are the contributions, conceptual--providing a conceptual framework that integrates several features of risk of exclusion--, methodological--developing a macro accounting approach that minimizes double counting--and empirical, by providing the first-ever global estimates of people at risk of exclusion. The revised abstract also expands on the main implications of our findings: While antipoverty policies can support household consumption and smooth its volatility among the poor, they are unlikely to address social exclusion stemming from ethnic, racial, or gender discrimination. Furthermore, interventions targeting discriminatory laws may not effectively address exclusion resulting from long-term unemployment. Therefore, addressing social exclusion is contextual, requiring a comprehensive suite of multiple interventions. All these issues have now been added to make more salient the research gap, the purpose of the paper, its key findings and the policy implications of such findings. 

Literature Review should be extensive with updated literature. Less use of older papers/literature. Theme wise presentation and sequencing should be checked.

The revised version of the literature review now brings more recent sources on exclusion, focusing on common blocks across studies that highlight issues associated with lack of participation in society, opportunity, ability and dignity, as well as new elements in the conceptual literature that emphasize a broader notion of social sustainability. The revised literature review also notes the emphases of different studies on outcomes and/or processes, and adds new views on exclusion more aligned with equity than participation. This gives a more diverse reading of the literature from the very start of the literature review. In reviewing the literature, we have kept a theoretical consistency and coherence, which is reflected in the conceptual framework, and to a lesser extent to the suggested sequencing of publications, which in our view is of a secondary importance.

Underpinning theory needs to be discussed.

Section 2 is explicitly devoted to discussing the underlying theories of social exclusion, which are captured by the conceptual framework that was developed for that purpose. The framework underscores the relative, multidimensional, agency and dynamic nature of exclusion as highlighted by theories of exclusion, as well as the dimensions, factors and drivers of exclusion and its social consequences. 

Proof reading required.

Please note that the revised version has been proofread by a professional editor and formatted according to the journal specifications.

Contribution of paper needs to be discussed.

In the revised manuscript, the abstract, introduction and conclusion sections explicitly discuss three specific contributions of the paper: conceptual, methodological and empirically. Conceptually, the paper develop an analytical framework that reflects the relative, dynamic and multidimensional features of exclusion and consider identity, circumstances and socioeconomic factors as the factors of exclusion. Methodologically, we develop a macro accounting approach that minimizes double counting of people at risk of exclusion across population groups. Empirically, we contribute to the literature by providing the first estimates populations at risk of social exclusion on a global and regional basis. The Analysis also provides exclusion estimates by country’s income level and fragility status. 

Research methods could be better explained with relevant citations of literature highlighting the use of particular method.

The methodology section is extensively explained in the paper and also in the supporting Information materials. We would like to highlight the fact that we offer the first ever methodology to measure populations at risk of exclusion based on a macro counting approach, so there is not comparable studies that use alternative methods for measure exclusion.

Responses to Reviewer 1

In figure 1, and 2 provide source. 

The source of Figure 1 and Figure 2 are the authors. We have included that information in the revised version of the manuscript.

In methodology section justify your formulas, which you used along with description. Description of variables and data table should be included. 

Regarding the justification of the formulas, for the sake of clarity and space we have move that discussion to the Supporting Information materials, in particular S1, section A, where the reader can find a full discussion on the mathematical justification of equation (3), the only one that is more elaborate. In the revised version, we have included a paragraph after the description of equation (3) to refers the reader to S1, section A for a more complete derivation and specification of the formula. 

Regarding the description of variables and data, we describe the variables in the data section and also in the Supporting Information section S3. 

Responses to Reviewer 2

Title: the current title needs to communicate the crux of the research and therefore the authors are suggested to rethink the title. The title should contain the purpose for this kind of study is useful, for instance, supporting global policy decisions on social inclusion in development.

We thank the reviewer for this suggestion but we feel the proposed title already communicates the crux of our research: we are contributing to the literature with new concepts, an innovative methodology that allow to measure populations at risk of social exclusion while minimizing double counting dimensions of exclusion, and the first ever estimates of populations at risk of social exclusion on a global and regional scale. The findings can indeed support several global policy areas. However, its main contributions are conceptual, methodological and empirical and that is captured in the current title.

Abstract : the abstract needs to be strengthened with inclusion of the key implications of the research in the concluding sentences.

Thanks for your comment. The revised abstract now highlight the research gaps and underscores the key contributions of the paper. The revised paper also emphasize the purpose of this paper, which is to improve the knowledge base about the global scale of social exclusion and its most likely sources. The abstract also calls the attention of the reader to our findings, namely, the number and share of people at risk of exclusion, and rates/shares by world regions and types of countries according to fragility and income levels. The revised abstract also enumerates the contributions of the paper in a more explicit manner. Three are the contributions, conceptual--providing a conceptual framework that integrates several features of risk of exclusion--, methodological--developing a macro accounting approach that minimizes double counting--and empirical, by providing the first-ever global estimates of people at risk of exclusion. The revised abstract also expands on the main implications of our findings: While antipoverty policies can support household consumption and smooth its volatility among the poor, they are unlikely to address social exclusion stemming from ethnic, racial, or gender discrimination. Furthermore, interventions targeting discriminatory laws may not effectively address exclusion resulting from long-term unemployment. Therefore, addressing social exclusion is contextual, requiring a comprehensive suite of multiple interventions. All these issues have now been added to make more salient the research gap, the purpose of the paper, its key findings and the policy implications of such findings. 

Conceptualizing social exclusion: In this section the authors can use a timeline to show how the concept of social exclusion has evolved over time.

The revised version brings more recent sources on exclusion, focusing on common blocks across studies that highlight issues associated with lack of participation in society, opportunity, ability and dignity, as well as new elements in the conceptual literature that emphasize a broader notion of social sustainability. The revised literature review also notes the emphases of different studies on outcomes and/or processes, and adds new views on exclusion more aligned with equity than participation. This gives a more diverse reading of the literature from the very start of the literature review. In reviewing the literature, we have kept a theoretical consistency, in light of the conceptual framework, and to a lesser extent a sequencing of the timing of the publications, which in our view is of a secondary importance.

Methodology: the authors should use a numerical example to explain the equations for comprehension of the readers.

We have included in the Methodology section as well as in the S1 section of the Supporting Information, several examples that explain all the equations presented in the paper.

Conclusion: the key points emerging from the analysis and how they relate to the wider context needs to be strengthened in this section.

The revised version of the conclusion section makes a clear and explicit point about the contributions of the paper, emphasizing the implications of the findings in terms of populations at risk of exclusion on a globally and regional scales. The conclusions also incorporate now a discussion of policy implications, encompassing three distinct points:

First, policies targeted at alleviating or ending most extreme forms of poverty might overlook those at risk of exclusion, which amount to about a fifth of the world’s population, or almost 1.5 billion people. Second, effectively addressing exclusion necessitates multiple interventions tailored to different groups and sustained over time. Third, the complexities in defining and measuring exclusion do not render it unmanageable. Nonetheless, effective addressing requires precise estimates and agreed-upon methodologies derived from available sources that can be frequently and easily monitored. Thus, we call for more and better-coordinated efforts measure, monitor and tackle the sources and drivers of social exclusion at a global level. We express our gratitude to Reviewer 2 for his/her comments. We believe that these have contributed to improve the final version of the paper. 

Other minor suggestions: Grammatical errors, and the errors in the references need to be corrected.

A professional copy editor has revised the revised version and eliminated any grammatical errors or typos that might have remained in the original version.

---

## [Decision Letter · Decision Letter 1]

18 Jan 2024

Social Exclusion: Concepts, Measurement, and a Global Estimate

PONE-D-23-14600R1

Dear Dr. Nino-Zarazua,

We’re pleased to inform you that your manuscript has been judged scientifically suitable for publication and will be formally accepted for publication once it meets all outstanding technical requirements.

Kind regards,

Avanti Dey, PhD

Staff Editor

PLOS ONE

Additional Editor Comments (optional):

Reviewers' comments:

Reviewer's Responses to Questions

**Comments to the Author**

1. If the authors have adequately addressed your comments raised in a previous round of review and you feel that this manuscript is now acceptable for publication, you may indicate that here to bypass the “Comments to the Author” section, enter your conflict of interest statement in the “Confidential to Editor” section, and submit your "Accept" recommendation.

Reviewer #1: All comments have been addressed

Reviewer #2: All comments have been addressed

2. Is the manuscript technically sound, and do the data support the conclusions?

Reviewer #1: Yes

Reviewer #2: Yes

3. Has the statistical analysis been performed appropriately and rigorously? 

Reviewer #1: Yes

Reviewer #2: Yes

4. Have the authors made all data underlying the findings in their manuscript fully available?

Reviewer #1: Yes

Reviewer #2: Yes

5. Is the manuscript presented in an intelligible fashion and written in standard English?

Reviewer #1: Yes

Reviewer #2: Yes

6. Review Comments to the Author

Reviewer #1: It's a very interesting paper, and it should be published.

Reviewer #2: No further comments to the authors. The authors have incorporated the suggestions in a comprehensive manner.

7. PLOS authors have the option to publish the peer review history of their article (what does this mean?). If published, this will include your full peer review and any attached files.

Reviewer #1: **Yes: **Dr Narendra N Dalei

Reviewer #2: **Yes: **Suparana Katyaini

---

## [Editor Report · Acceptance letter]

19 Feb 2024

PONE-D-23-14600R1 

PLOS ONE

Dear Dr. Niño-Zarazúa, 

I'm pleased to inform you that your manuscript has been deemed suitable for publication in PLOS ONE. Congratulations! Your manuscript is now being handed over to our production team.

Kind regards, 

on behalf of

Dr. Avanti Dey 

Staff Editor

PLOS ONE